# Wear Mechanism of Multilayer Coated Carbide Cutting Tool in the Milling Process of AISI 4340 under Cryogenic Environment

**DOI:** 10.3390/ma15020524

**Published:** 2022-01-11

**Authors:** Shalina Sheik Muhamad, Jaharah A. Ghani, Che Hassan Che Haron, Hafizal Yazid

**Affiliations:** 1Department of Mechanical and Manufacturing Engineering, Faculty of Engineering and Built Environment, Universiti Kebangsaan Malaysia, Bangi 43600, Selangor, Malaysia; shalina@nm.gov.my (S.S.M.); chase@ukm.edu.my (C.H.C.H.); 2Prototype and Plant Development Center (PDC), Technical Support Division, Malaysian Nuclear Agency, Bangi 43000, Selangor, Malaysia; 3Materials Technology Group (MTEG), Industrial Technology Division, Malaysian Nuclear Agency, Bangi 43000, Selangor, Malaysia; hafizal@nm.gov.my

**Keywords:** cryogenic cutting, AISI 4340, coated carbide cutting tool, wear mechanisms, temperature distribution

## Abstract

Cryogenic technique is the use of a cryogenic medium as a coolant in machining operations. Commonly used cryogens are liquid nitrogen (LN2) and carbon dioxide (CO_2_) because of their low cost and non-harmful environmental impact. In this study, the effects of machining conditions and parameters on the wear mechanism were analysed in the milling process of AISI 4340 steel (32 HRC) under cryogenic conditions using a multilayer coated carbide cutting tool (TiAlN/AlCrN). A field emission scanning electron microscope with energy-dispersive X-ray analysis was used to examine the wear mechanisms comprehensively. At low machining parameters, abrasion and adhesion were the major wear mechanisms which occurred on the rake face. Machining at high machining parameters caused the removal of the coating material on the rake face due to the high temperature and cutting force generated during the cutting process. In addition, it was found that continuously adhered material on the rake face would lead to crater wear. Furthermore, the phenomenon of oxidation was also observed when machining at high cutting speed, which resulted in diffusion wear and increase in the crater wear. Based on the relationship between the cutting force and cutting temperature, it can be concluded that these machining outputs are significant in affecting the progression of tool wear rate, and tool wear mechanism in the machining of AISI 4340 alloy steel.

## 1. Introduction

AISI 4340 steel material is widely used in the aerospace industry for aircraft landing gear, power transmission gears and shafts in the automotive industry, as well as for other structural components. Manufacturing processes in the automotive and aerospace industries require optimal dimensional accuracy and geometrical characteristics. As a result, poor machinability, such as the use of worn or damaged tools, may result in a loss of quality of the finished part. The alloying element in AISI 4340 provides a balance of strength, wear resistance and durability compared to normal steel [1]. Tool wear is related to the speed of cutting. It has been reported in the literature that a cutting speed of up to 250 m/min can be achieved by using coated carbide tools and a maximum of 420 m/min can be achieved by using ceramic tools [2]. Physical Vapour Deposition (PVD)-coated carbide, Chemical Vapour Deposition (CVD)-coated carbide and ceramic tools are the most commonly used coated tools. The primary purpose of the coating is to provide mechanical, chemical and thermal protection at higher temperatures. 

In a previous study, Li et al. [3] used three types of coating materials (TiAlN+TiN, AlTiN and TiN/TiCN/TiAlN multi-coated) during high-speed milling of AISI 4340 to investigate the surface roughness, cutting force and tool wear aspects. The results from this investigation showed lower surface roughness, minimal flank wear width and the smallest resultant cutting force were generated by the use of TiN/TiCN/TiAlN multi-coated tools. The wear resistance and wear mechanism of the TiN/TiCN/TiAlN multi-coated tools were better than that of the AlTiN and TiAlN+TiN coated tools due to the coating materials and structure. 

In the case of AISI 4340, its martensitic structure and chemical composition will influence the mechanism of wear of the cutting tools [4,5]. According to Chinchanikar and Choudhury [6] during the machining of AISI 3430 with coated PVD and CVD tools, adhesion and abrasion, flank wear and flaking at the clearance face and nose were dominant when using CVD-coated tools. The main forms and mechanisms of wear for PVD-coated tools, were abrasion, adhesion, diffusion and crater wear. Plastic deformation was detected at the end life of both coated tools. Boing et al. [7] had also reported similar findings on the mechanisms of wear. The wear mechanisms of the CVD-coated tool was related to abrasion, break and dissemination, fracture, peeling, separation amongst layers and fragments breaking off. PVD coating wear mechanisms involved abrasion, high deformation leading to crack nucleation of the coating at the edge which reduced the binding power of the tool substrate and coating, and caused fragments to be broken off from the edge.

In order to overcome the problems of increased temperature and tool wear during machining by means of a heat mechanism, cutting fluids are used, mostly by flooding the cutting area. Techniques such as this have proven to be unsustainable and harmful to the environment. Normally, cutting fluids must be treated and disposed of at the end of their useful lives. It is vital that an efficient disposal system is implemented. Cutting fluids also cause health and safety risks to workers when in contact with them [8,9]. Costs related to the consumption of cutting fluids can be up to 17% to 30% of the overall cost of the machining [10]. Disposal and chemical treatment of cutting fluids contribute half of the cost of machining [11].

In recent years, researchers have explored techniques which are more efficient and sustainable. Among the sustainable techniques in machining is cryogenic cooling [8]. This technique is considered a sustainable technique as it complies with the three pillars of sustainable manufacturing, which are economic, societal and environmental [12]. Liquid nitrogen (LN2) and carbon dioxide (CO_2_) are commonly used as a coolant [13]. The coolant is environmentally friendly without harmful or toxic effects. Significant research on cryogenic cooling has focused on alternative sustainable strategies and has significantly improved the machining performance. The machining performance of AISI 4340 material is improved during machining by the use of cryogenic cooling [14]. It was reported that cryogenic cooling lowers the temperature (32% to 33%) in contrast to dry machining. LN2 reduces the contact length of the chip-tool and, therefore, reduces the tool wear, particularly crater wear. LN2 could reduce machining forces by 15% to 26% and feed forces by 54% to 56% in simulations and experiments. Halim et al. [15] carried out research on tool life, tool wear patterns, wear mechanisms and chip morphology using dry cutting as well as under cryogenic carbon dioxide (CO_2_) conditions. The tool life was improved by up to 70.8% and the machining temperature reduced up to 80% in comparison to dry conditions. According to Musfirah et al. [16], some research has been carried out on cryogenic machining, but few investigations related to the milling operation [16]. Furthermore, the tool life was shorter in cryogenic LN2 compared to dry cutting in the milling process. In addition, the tool undergoes large temperature variations in each revolution of milling operation. This is attributed to the interrupted cutting process caused by the drastic temperature changes due to the use of LN2, which induces the tool to contract and expand repeatedly, culminating in fatigue and cracking.

The main objective of this study was to examine the effect of cryogenic LN2 during end milling of AISI 4340 using a multi-coated insert. Machining parameters (cutting speed, feed rate, axial depth of cut and radial depth of cut) were varied during the experiments. The analysis was focused on the tool wear mechanism and its relationship with the tool life, cutting force, and the cutting temperature.

## 2. Materials and Methods

A rectangular block of 165 × 100 × 100 mm, made of AISI 4340 alloy steel, was used as a material workpiece. The chemical properties of the material workpiece are shown in Table 1. The workpiece’s surface hardness was 32 HRC. In the experiments, a PVD grade ACK 300 cutting tool (Sumitomo, Tokyo, Japan) insert with multi-coated alternate layers of TiAlN and AlCrN with a coating thickness of 3 μm was used. The toolholder was a Sumitomo WEX 2020E end milling cutter (Sumitomo, Tokyo, Japan) with a nominal diameter of 20 mm. The cutting tool holder and end mill insert geometry used for this research study are shown in Table 2. Nine experiments were conducted under cryogenic LN2 conditions, based on the cutting parameters shown in Table 3. The variables included the cutting speed, feed rate, axial and radial depth of cut. The experiments were conducted based on the Taguchi Method L9 orthogonal array as in Table 4.

Figure 1a shows the multilayer coated carbide cutting tool dimensions. The coolant used was liquid nitrogen channeled through a nozzle directly into the cutting zone during the milling process as can be seen in Figure 1b. A nozzle with an inner diameter of 3.5 mm was placed at an angle of 45° towards the tip of the cutting tool and a spray distance of 50 mm from the cutting tool rake face. A flow rate was maintained at 0.001159 m^3^ /s during the machining process.

The tool life criteria for the experiment were established on the ISO-8688, [19] (average wear = 0.3 mm or maximum wear = 0.5 mm, whichever occurred first). A field emission scanning electron microscope (FESEM, Zeiss model Gemini SEM 500, Zeiss, Jerman) with energy dispersive X-ray spectroscopy (EDAX, PANalytical model X’Pert PRO, UK) was used to analyse the wear mechanism and elements on the worn tool surfaces. The cutting force was measured using an in-house developed NeoMoMac dynamometer (Universiti Kebangsaan Malaysia, Bangi, Malaysia).

## 3. Results and Discussion

### 3.1. Tool Life under Cryogenic Conditions

The tool life obtained after milling AISI 4340 under cryogenic conditions is shown in Figure 2. Under this cryogenic condition, the maximum machining time was 45 min in Experiment 1 (Vc: 200 m/min, fz: 0.15 mm/tooth, ap: 0.3 mm and ae: 0.2 mm). Experiment 6 (Vc: 250 m/min, fz: 0.30 mm/tooth, ap: 0.4 mm and ae: 0.2 mm) and 9 (Vc: 300 m/min, fz: 0.30 mm/tooth, ap: 0.3 mm and ae: 0.35 mm) achieved a minimum tool life of approximately 21 min. The dominant variables of cutting tool life depended on the cutting speed and feed rate, where minimum machining parameter combinations resulted in the lowest tool wear progression, as in Experiment 1. This finding was in agreement with the findings of Halim et al. [15] who reported that longer tool life was observed at minimum cutting speed values.

Figure 2 shows the relationship between progress of tool flank wear and cutting time during cryogenic end milling. Generally, the progress of flank wear can be divided into three segments: the initial stage, the steady state stage and the final stage. The progression of wear increases abruptly at the early stage and remains stable after 0.125 mm. The progress rose steadily until the life of the cutting tool was reached. It appears from Figure 2 that the growth of flank wear increased gradually due to good temperature control under cryogenic conditions. This was in line with Strano et al. [20] who pointed out that longer tool life was observed as cutting speed increased under cryogenic cooling. Kaynak and Gharibi [21] reported a 25% reduction in flank wear rate of AISI 4140 in a cryogenic LN2 cutting environment as compared with dry cutting. As the cutting process was carried out at extremely low temperatures, the tool wear over time was reduced compared to dry machining because the cutting tool material hardness could be maintained [22]. As a result, it delayed the rate of wear and prolonged cutting tool life. However, this finding differs from Shokrani et al. [23], who reported that longer tool life was observed in dry cutting compared to cryogenic cutting using LN2. As reported, the cryogenic cooling increases the material strength and hardness. It can be concluded that the interrupted milling process, caused the tool to contract and expand repeatedly, leading to chipping.

The tool life and cutting length for nine experiments of multilayer coated carbide cutting tool is shown in Table 5. The cutting length is the length of cuts when the cutting tool met the tool life criteria. Table 5 shows that the maximum cutting length was obtained in Experiment 9. Meanwhile, the minimum of 22.28 m was achieved in Experiment 1. The results indicate that the use of cryogenic coolants greatly prolong the lifespan of a tool, which is in line with the results in the literature [15]. The results reported here appear to support the assumption that cryogenic LN2 dissipates heat and reduces cutting temperature at high cutting speeds of milling. Furthermore, according to Halim et.al [15], cryogenic coolants are more effective at high cutting speed compared to low cutting speed.

### 3.2. Tool Wear and Wear Mechanisms of a Multilayer Coated Carbide Cutting Tool

FESEM photomicrographs and EDAX spectrum were taken from the worn cutting tools, wherein the tools met the tool life criteria. Figure 3 shows the resulting wear mechanism for cryogenic milling. Nose wear was formed at the respective areas as shown in Figure 3a, crater wear was identified as forming at medium cutting speed, while flaking occurred at high cutting speed. From Figure 3, during the machining of AISI 4340 the crater wear had predominantly occurred on the rake face. This wear was largely dependent on the hardness of the material, machining parameters, cutting tool geometry and cutting fluid [24]. Tang et al. [25] pointed out that crater wear which had been initiated by high heat and mechanical loads, resulting in higher compressive stress and shear stress, occurred at the cutting tool–chip interface and resulted from friction effects. The growth of crater wear was speeded up by the existence of abrasion and adhesion on the tool rake face, as can be seen in Figure 3b,c. This wear is common during the machining of hardened steels, due to hard carbides present in the microstructure of the material. The rubbing between the tool, chips and workpiece which occurred on the tool flank face during the milling of AISI 4340 led to larger abrasion wear. Rigorous chipping of the coating layer and notching on the rake face were observed in Experiment 9 (Vc: 300 m/min, fz: 0.30 mm/tooth, ap: 0.3 mm and ae: 0.35 mm) as can be seen in Figure 3c. This phenomenon occurred due to the high feed rate, which resulted in increased cutting temperature and greater cutting force. Although notching and flaking were observed, there was no fracture of the cutting insert.

Adhered material was observed on the tool rake face as can be observed in Figure 4. The EDAX spectrum on the area confirmed the adherence of the workpiece material (Fe element) and the TiAlN/ AlCrN coating on the surface of the tool. Peeling off the coating would result in the uncovering of the carbide tool substrate material. Typically, the attachment of the material workpiece on the flank and rake face is the result of the extreme stresses and high temperature at the interfaces [26]. Continued adherence occurred on the newly generated surface and the adhered layer surface, causing the layer thickness to increase. During the chip flow or movement of the workpiece, the layer was pulled by force. Thus, the cutting tool area would be exposed due to the pull-outs. The repeated process of this formation would lead to crater wear. This is in line with Musfirah [16] who ascribed the disposition of BUE at the tool edge as the adhesive wear mechanism. BUE formation at the tool edge is due to the chemical bonding between the material substrate of the tool and the workpiece. In this situation, machining at high speed and AISI 4340 toughness factors made the tool susceptible to adhesion.

Figure 5 shows the occurrence of adhesion wear on the tool flank face. It was evident that the carbide particles in the AISI 4340 workpiece were the cause of grooves. The workpiece material (Fe, Mn, C) and the coating tools material (Al, Cr) were detected by analysis of the chemical element (Figure 6). In addition, oxygen was also detected. This result showed that there was an oxidizing reaction. Furthermore, the high W elements detected indicated that the tool coating layer had been peeled off, uncovering the carbide material tool substrate. Without the multi-coated layers, the flaking on the rake’s face was enlarged and the wear progressed rapidly. If the machining process were continued, it would lead to an enlargement of flaking due to continuous mechanical and thermal load in the intermittent milling process.

Carbide particles such as iron-carbide (Fe3C), vanadium-carbide (VC), molybdenum-carbide (Mo2C) and chromium-carbide (Cr7C3) present in AISI 4340 which are highly abrasive ultra-hard particles would vigorously rub between the machined surface and tool flank, resulting in abrasion marks [6,27]. From the FESEM images (Figure 6), the existence of Fe, Mn, C elements from the AISI 4340 workpiece can be seen. Carbide elements tend to rub the tool surface during the cutting process, leading to greater abrasive wear. In the multi-coated tools, the abrasive wear mechanism resulted in the delamination of the coating which exposed the tools’ base material. These results were similar to those obtained by Chinchanikar and Choudhury [28] for dry machining of AISI 4340 with a coated multilayer coated carbide cutting tool. Their studies revealed that the presence of hard carbides and cementite in steels was the reason for abrasion wear. This would also cause the particles to break away from the tool cutting edge, resulting in more serious damage.

### 3.3. Effects of Force and Cutting Temperature on Cutting Tool Wear

Evaluation of the cutting force components during the machining process is one of the significant measures of machining performance. Figure 7 shows the resultant force under cryogenic cooling conditions for nine experiments. The resultant force (total cutting force) was calculated using Equation (1) [29]
(1)$$F_{r}=\sqrt{F_{x}{}^{2}+F_{y}{}^{2}+F_{z}{}^{2}}$$

The data on cutting force was extracted at the start of the cutting process to eradicate the tool wear effect. Measurements of the components of the cutting force were obtained thrice at an average wear = 0.1 mm. The average of these values was then determined for each force component (*F_x_, F_y_* and *F_z_*). Previous studies by Muhamad [30] highlighted machining parameters associated with cutting force during cryogenic LN2 machining. Cryogenic milling with low cutting temperatures had successfully reduced the cutting force. In their comprehensive investigation, Varghese et al. [31] concluded that besides decreasing cutting force, cryogenic machining also reduces the welding and adhesion of chips on the machine surface, leading to stable machining conditions. In addition, the cryogenic condition also creates the formation of a pad between the tool and the workpiece which enhances greasing and lessens the chatter of the cutting tool.

Figure 7 shows the cutting forces and cutting temperatures generated in the experiments. It clearly shows that the cutting temperature increases with cutting force. Also, as can be seen in Figure 7, the effect of resultant force for Experiments 1, 6 and 9 correlates with Figure 3 (tool wear in rake face). The generated force (Figure 7) also tends to be associated with Figure 3. Serious flaking and diffusion wear did exist in Experiment 9 which was the result of the much larger thermo-mechanical impact on tool-chip boundaries which had probably been caused by excessive cutting force. It showed that the abrasion wear observed in tool 6 was less than in tool 9, although the cutting tool life had been achieved in equal time. Higher cutting forces subsequently caused greater temperatures in the zone of cutting, which contributed significantly to dimension precision, surface quality and tool life [32].

The cutting temperatures at the cutting edge area of the tool have an effect on the wear rate of the cutting tool and on the friction between the tool and chip. This generated temperature could affect the cutting tool life and cutting force [33]. The temperature distributions on the cutting area were estimated using the Thirdwave AdvantEdge commercial finite element analysis software. It was found that there was an extreme temperature gradient in the middle tool-chip contact area on the rake face as shown in Figure 8. Machining with cryogenic LN2 at Vc: 200 m/min, fz: 0.15 mm/tooth, ap: 0.3 mm and ae: 0.2 mm generated 576 °C, while the maximum simulated temperature was 850 °C when machining at Vc: 300 m/min, fz: 0.30 mm/tooth, ap: 0.3 mm and ae: 0.35 mm. At a temperature of more than 800 °C on the contact area between the tool and the chip, the tool material will diffuse beyond the interface [34]. Energy-dispersive X-ray (EDAX) analysis highlighted that the AISI 4340 workpiece elements existed in the flank face as can be seen in Figure 5 and Figure 6. High cutting temperature causes adhesive and diffusion wear and enhances the crater wear and, as a result causes more degeneration of the cutting tool material [25]. This shows that cryogenic machining was efficient in lowering the temperature of the machining process and decreased the chemical affinity of the tool and workpiece.

Jebaraj et al. [35] had observed significant decreases in temperature after end milling operations of 55NiCrMoV7 die steel using LN2, compared to CO2 and commercial cutting fluids. In this investigation, the reduction in temperature and force, which reduces the thermal effects on the cutting tool, reduced abrasion and diffusion wear mechanisms resulting in improved cutting tool life. The temperature at the tool-chip interface during cryogenic cutting was raised by an increase in the feed rate. The rise in temperature with an increase in feed rate is due to the increase in friction [36].

## 4. Conclusions

The effect of the cryogenic LN2 technique on milling of AISI 4340, was investigated using multilayer coated carbide cutting tool (TiAlN/AlCrN). The tool life, wear mechanisms, cutting force and temperature were evaluated. The following conclusions have been drawn:The use of a cryogenic coolant plays an important role in reducing the progress of tool wear. Consequently, milling under cryogenic conditions increases the length of cutting and prolongs the tool life.The main wear mechanism of the multilayer coated carbide cutting tool were abrasion, adhesion and diffusion on the rake face during cryogenic milling of AISI 4340. From the FESEM micrograph shown, the topography included adhered chip particles, grooves, BUE and rough surfaces on the cutting tools.The mixture of abrasion, adhesion and diffusion wear accounted for the crater wear of the carbide coated tool in Experiment 9 when compared to Experiment 1 because the cutting edge experienced higher temperatures, more friction and more tearing due to excessive cutting force.The application of the cryogenic technique is recommended as it leads to a sustainable manufacturing process, with only a small amount of LN2 dissipating to the atmosphere, which does not cause environmental pollution.

## Figures and Tables

**Figure 1 materials-15-00524-f001:**
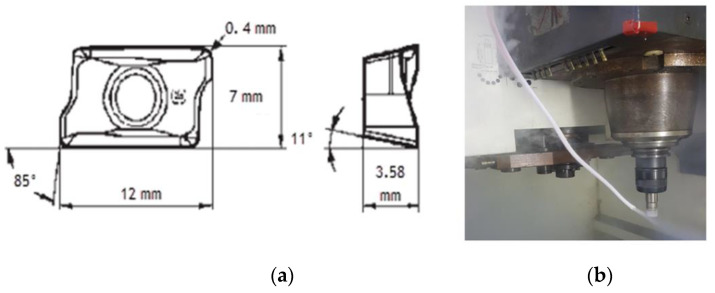
(**a**) The multilayer coated carbide cutting tool; (**b**) LN2 delivery.

**Figure 2 materials-15-00524-f002:**
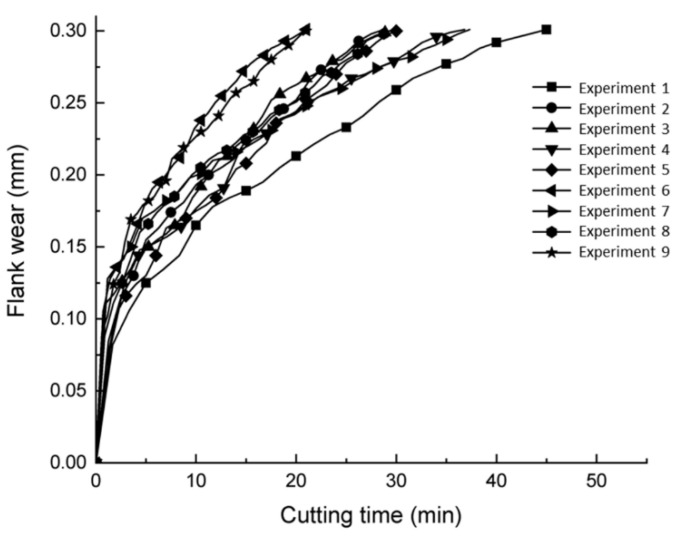
Flank wear vs. cutting time under cryogenic cooling conditions.

**Figure 3 materials-15-00524-f003:**
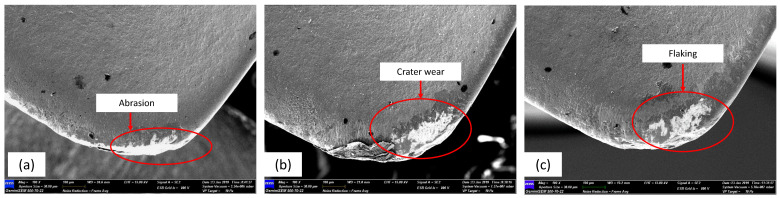
Tool wear in rake face: (**a**) Abrasion wear in Experiment 1; (**b**) Crater wear in Experiment 6; (**c**) Flaking in Experiment 9.

**Figure 4 materials-15-00524-f004:**
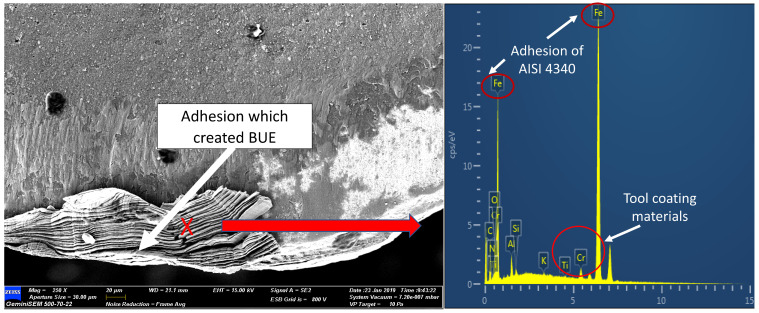
Adhesion wear on the rake face and EDAX analysis: Experiment 6.

**Figure 5 materials-15-00524-f005:**
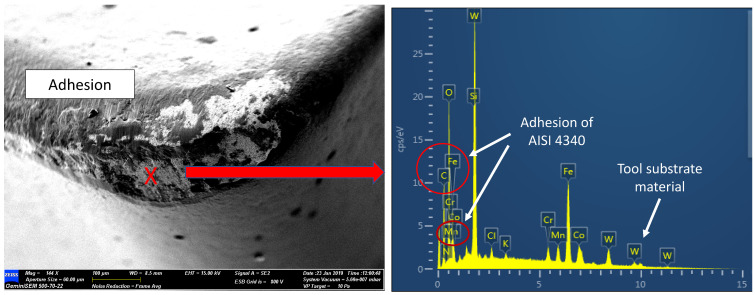
Adhesion wear on the flank face and EDAX analysis: Experiment 9.

**Figure 6 materials-15-00524-f006:**
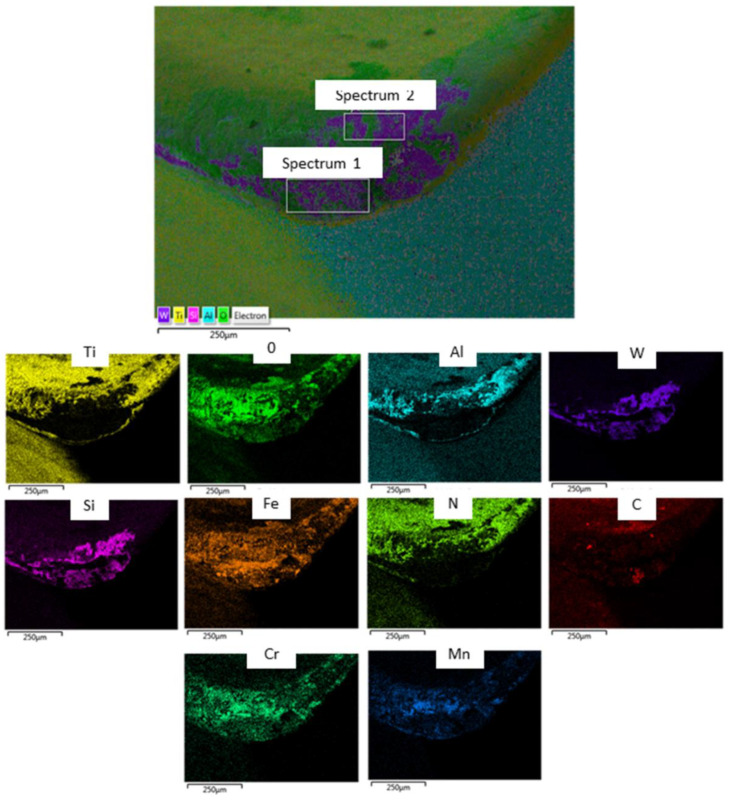
FESEM analysis which confirmed the existence of Fe, Mn, C particles in AISI 4340: Experiment 9.

**Figure 7 materials-15-00524-f007:**
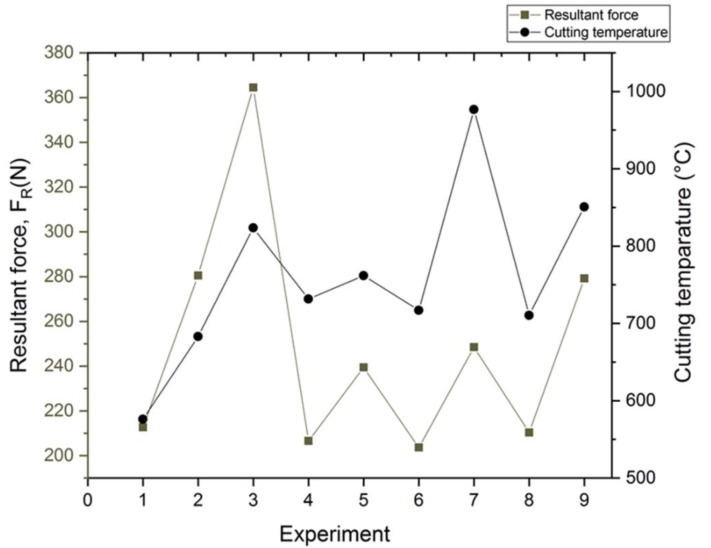
The resultant force and cutting temperature for the nine experiments under cryogenic milling.

**Figure 8 materials-15-00524-f008:**
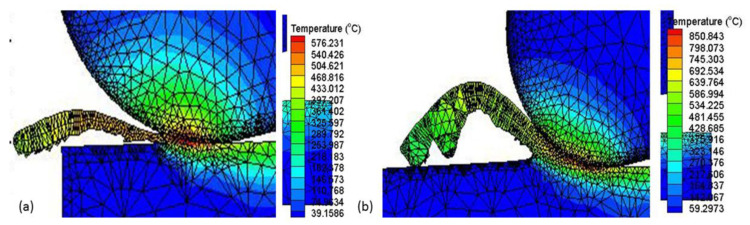
The simulated temperature distributions on the rake face: (**a**) Experiment 1; (**b**). Experiment 9.

**Table 1 materials-15-00524-t001:** AISI 4340 composition (wt.%) [17].

Element	C	Si	Mn	P	S	Cr	Mo	Ni
Minimum (%)	0.37	0.10	0.55	0.00	0.00	0.65	0.2	1.55
Maximum (%)	0.44	0.35	0.90	0.04	0.04	0.95	0.35	2.00

**Table 2 materials-15-00524-t002:** Cutting tool and end mill insert geometry [18].

Items	Value
Number of inserts	3
Tool diameter	20 mm
Rake angle	28°
Clearance angle	11°
Coating material	TiAlN /AlCrN

**Table 3 materials-15-00524-t003:** Cutting conditions.

Machine centre	DMC 635 V Eco
Workpiece material	AISI 4340
Cutting speed, Vc (m/ min)	200, 250, 300
Feed rate, fz (mm/tooth)	0.15, 0.20, 0.30
Axial depth of cut, ap (mm)	0.3, 0.4, 0.5
Radial depth of cut, ae (mm)	0.20, 0.35, 0.50
Cutting configuration	Down milling
Lubricant	Cryogenic LN2 with flow rate of 1.159 × 10^−3^ m^3^/s

**Table 4 materials-15-00524-t004:** Taguchi Method L9 Orthogonal Array design of experiments.

Exp. No.	vc (m/min)	fz (mm/tooth)	ap (mm)	ae (mm)
1	200	0.15	0.3	0.20
2	200	0.20	0.4	0.35
3	200	0.30	0.5	0.50
4	250	0.15	0.5	0.35
5	250	0.20	0.3	0.50
6	250	0.30	0.4	0.20
7	300	0.15	0.4	0.50
8	300	0.20	0.5	0.20
9	300	0.30	0.3	0.35

**Table 5 materials-15-00524-t005:** Tool life and cutting length of multilayer coated carbide cutting tools.

Experiment	Tool Life (mins)	Cutting Length (m)
1	45.0	22.28
2	28.3	18.98
3	29.8	28.05
4	36.8	21.45
5	30.0	24.75
6	21.0	24.75
7	37.3	26.40
8	29.6	28.05
9	21.0	29.70

## Data Availability

Not applicable.

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
