# Peer review of "Wear Mechanism of Multilayer Coated Carbide Cutting Tool in the Milling Process of AISI 4340 under Cryogenic Environment"

_materials, 2022, doi:10.3390/ma15020524_

Round 1
Reviewer 1 Report
The author performed the critically the wear mechanism of multilayer coated carbide insert in the milling process of AISI 4340 steel under cryogenic condition. It was concluded that responses are significant in affecting the progression of tool wear rate. The article is interesting.
- Speed of cutting should be replaced with cutting speed.
- Machining speed should be replaced with cutting speed.
- Add more write up related to Ref. [11] in introduction section.
- Remove word ‘They’.
- A nozzle with an inner diameter of 3.5 mm was 116 placed at an angle of 45 ° towards the tip of the cutting tool and a spray distance of 50 mm 117 from the cutting tool rake face. A flow rate was maintained at 0.001159 m3 /s during the 118 machining process. Any reference ?
- On what basis input parameters have been chosen?
- How cutting length is determined?
8.Exerimental set up figure is missing.
Author Response
The reviewers comments already been addressed in the uploaded file.

Reviewer 2 Report
The article is very interesting.
Additional questions that may arise include:
1. Was the quality of the obtained surface after processing measured?
2. How was the temperature measured during the milling?
3. Has the chip upset been measured and the results compared with the conventional process and with cryogenic cooling? Information about surface roughness and upset can tell us more about the process.
4. In what program was the simulation carried out?
Author Response
The reviewer comments already been addressed in the upload file.

Reviewer 3 Report
The work of this manuscript seems a little interesting and potential application. The authors mainly investigated wear mechanism of multilayer coated carbide cutting tool under cryogenic environment. It was proposed that the use of a cryogenic coolant plays an important role in reducing the progress of tool wear. The work is relevant and meaningful in helping our understanding of the extending life of the cutting tools. However, there are still some issues in the current manuscript. The authors need to deeply revise the manuscript to make it more readable. The manuscript can be probably considered for publication after the revision.
1) In the section of “2. Materials and Methods”, night sets of the experiment were carried out under different parameters (there are four different parameters with each of three values. Why were night sets of the experiment selected? Except for the three sets of Experiments 1, 6 and 9, The parameters of other sets were not clearly given.
2) In the Table 3, three different milling speeds (200, 250 and 300 m/min) were selected. However, the differences in three milling speeds are not very big, why did you select these three milling speeds?
3) In the section of “3. Results and discussion”, there are a lot of discussion about the extension of tool life in cryogenic environment. However, this manuscript did not carry out extra comparative experiments.
4) Figure 7 shows the variation in resultant force and cutting temperature. The resultant forces of Experiment 5 and Experiment 7 are relatively close, but the cutting temperatures differ greatly. can the author explain this phenomenon?
5) The study of the manuscript is not rigorous. For instance, variances of some results are not marked as shown in Figures 7.
6) There are major issues in the format and details of the manuscript as a whole, and the authors needs to proofread it. For instance, format of the references is not rigorous, and Figure 8(b) lacks description.
Author Response

(The authors gave the same response as above.)

Round 2
Reviewer 3 Report
My comments have been responded, and it is recommended to be accepted.